# Twenty Years of Anti-Vascular Endothelial Growth Factor Therapeutics in Neovascular Age-Related Macular Degeneration Treatment

**DOI:** 10.3390/ijms241613004

**Published:** 2023-08-21

**Authors:** Bo-Hyun Moon, Younghwa Kim, Soo-Young Kim

**Affiliations:** 1Department of Oncology and Lombardi Comprehensive Cancer Center, Georgetown University Medical Center, Washington, DC 20057, USA; bm366@georgetown.edu; 2Department of Paramedicine, Kyungil University, Gyeongsan-si 38428, Gyeongbuk, Republic of Korea; 3Department of Pharmaceutics, Virginia Commonwealth University, Richmond, VA 23298, USA

**Keywords:** retina, neovascularization, choroid, retinal pigment epithelium

## Abstract

Neovascular age-related macular degeneration (nAMD) is the primary disastrous retinal disease that leads to blindness in the elderly population. In the early 2000s, nAMD resulted in irreversible vision loss and blindness with no available treatment options. However, there have been breakthrough advances in the drug development of anti-angiogenic biological agents over the last two decades. The primary target molecule for treating nAMD is the vascular endothelial growth factor (VEGF), and there are currently several anti-VEGF drugs such as bevacizumab, ranibizumab, and aflibercept, which have made nAMD more manageable than before, thus preventing vision loss. Nevertheless, it should be noted that these anti-VEGF drugs for nAMD treatment are not effective in more than half of the patients, and even those who initially gain visual improvements lose their vision over time, along with potential deterioration in the geography of atrophy. As a result, there have been continuous endeavors to improve anti-VEGF agents through better efficacy, fewer doses, expanded intervals, and additional targets. This review describes past and current anti-VEGF therapeutics used to treat nAMD and outlines future directions to improve the effectiveness and safety of anti-VEGF agents.

## 1. Introduction

Age-related macular degeneration (AMD) is a complex chronic inflammatory retinal disease influenced by aging, environmental, and genetic factors. AMD is characterized by the accumulation of drusen and abnormal pigmentation. AMD is advanced to the late stages of geographic atrophy (GA) and/or choroidal neovascularization (CNV). Currently, there are age-related eye disease studies (AREDS), dietary supplements for early and intermediate AMD, anti-complement pathway drugs such as pegcetacoplan (Syfovre; Appellis Pharmaceuticals, Waltham, MA, USA) and avacincaptad pegol (Izervay; Iveric Bio, Cranbury, NJ, USA) for GA, and several anti-vascular endothelial growth factor (VEGF) drugs such as bevacizumab (Avastin; Genentech, San Francisco, CA, USA), ranibizumab (Lucentis, Genentech), and aflibercept (Eylea; Regeneron, Tarrytown, NY, USA) for the treatment of neovascular AMD (nAMD) and CNV. Whereas anti-complement pathway drugs were recently approved in 2022 and 2023 to treat GA and still require real-world post-market proof of their effectiveness and safety, anti-VEGF drugs have already shown quite successful efficacy for the last 20 years. However, despite the success of these anti-VEGF agents in treating nAMD, more than half of CNV patients are unresponsive to anti-VEGF agents [1,2], and the repeated ocular treatment of every or every other month schedule could be a burden to elderly patients. Furthermore, recurrent CNV can occur if patients fail to adhere to frequent injections. Intravitreal (IVT) injection, used in current treatment, has the potential to induce endophthalmitis, cataracts, hemorrhage, and retinal detachment. Continuous attempts to improve anti-VEGF agents have led to the development of new drugs such as Susvimo (ranibizumab ocular implant, Genentech) and Vabysmo (faricimab, anti-VEGF, and anti-angiopoietin2 (Ang2); Genentech) with increased extended duration or bispecific targets. The journey to improve anti-VEGF agents is constantly leading to new kinds of drugs featuring diversified target specificities, improved delivery systems, and target protein degradation. In this review, we describe the molecular structure of VEGF and the receptor binding, as well as the molecular and clinical aspects of existing and newly approved commercial anti-VEGF agents, clinical-stage-developing anti-VEGF agents, and further future directions on nAMD drug development.

## 2. Vascular Endothelial Growth Factor Family

VEGFs play an important role in vascular development and choroid maintenance in the normal eye. The basolateral secretion of VEGF from the retinal pigment epithelium (RPE) continues throughout life and mediates RPE survival [3]. However, the increase in VEGF secretion from RPE and the loss of RPE polarity are causes of the pathologic CNV condition [4]. The VEGF family in humans consists of five ligand members: VEGFA, VEGFB, VEGFC, VEGFD, and placenta growth factor (PGF), and there are three receptors: VEGFR1 (Fms-like tyrosine kinase 1, Fit-1), VEGFR2 (kinase insert domain receptor, KDR), and VEGFR3 (Flt-4) [5]. VEGFA binds to both VEGFR1 and VEGFR2, while VEGFB and PGF bind to VEGFR1 [6]. On the other hand, VEGFC and VEGFD bind with high affinity to VEGFR3 (Figure 1A) [7]. Although VEGFR1 has a high affinity with VEGFA, its weak tyrosine autophosphorylation suggests that the endothelial proliferation of VEGFA is primarily mediated via VEGFR2 [8]. VEGFR1 is even considered an antagonist against the VEGFA/VEGFR2 signaling pathway due to its weak receptor response. VEGFR2 is the dominant auto-phosphorylated tyrosine kinase receptor for VEGFA ligand binding. VEGFA/VEGFR2 signaling is majorly associated with endothelial survival and angiogenesis [9]. However, there is more complexity in VEGF and VEGFR signaling: the existence of soluble forms of VEGFRs, receptor heterodimerization (R1/R2), receptor dimerization between normal and soluble receptor forms (Figure 1A,B), and even ligand heterodimerization [5,10,11]. These all make it more elusive to understand VEGF and VEGFR signaling and require further studies and understanding of them in nAMD pathological conditions that are progressing.

VEGFA has variant isoforms ranging from 121 amino acids (a.a) to 206 a.a in length: VEGFA121, VEGFA145, VEGFA165, VEGFA183, VEGFA189, and VEGFA206 (Figure 2A). Notably, VEGFA121 is the most diffusible form, lacking heparin- and neuropilin (Nrp)-binding domains, whereas VEGFA189 and VEGF206 bind to the cell surface and extracellular matrix with high affinity with heparin and Nrp. VEGFA165, which has a low affinity to bind heparin and Nrp [5], is the predominant isoform of VEGFA in the adult retina and choroid [3]. It is further noteworthy that the type b forms, described as VEGFAxxxb such as VEGFA121b and VEGFA165b, are derived from distal spliced exon 8 (Figure 2B), possess distinct N-terminal amino acids, ending with SLTRKD instead of CDKPRR, and have anti-angiogenic function [12,13]. VEGFAxxxbs seem to bind to VEGFRs with similar affinities as VEGFAxxx and even activate VEGFR signaling pathways, albeit to a lesser extent. They are even considered inhibitory isoforms of VEGFA due to their lower receptor autophosphorylation [14,15]. However, their physiological function and relevant role as a drug development candidate still remain to be elucidated.

Finally, a comprehensive understanding of VEGFA isoform expression, VEGFA dimerization, ligand/receptor binding affinities, receptor dimerization, autophosphorylation, and stimulation rates associated with disease pathogenesis may lead to improvements in existent anti-VEGF drugs and/or new kinds of anti-angiogenic drug development.

## 3. Molecular and Clinical Aspects of Anti-VEGF Drugs

Most anti-VEGF agents developed and used for nAMD treatment are derived from antibodies or Fc-fused proteins (FcFPs) (Figure 3 and Table 1). Bevacizumab, ranibizumab, brolucizumab (Beovu; Norvatis, Basel, Swizerland), and faricimab are derived from humanized or primate antibodies. Bevacizumab is a monoclonal antibody (mAb). Ranibizumab is an antigen-binding fragment (Fab), brolucizumab is a single-chain fragment variable (scFv), and faricimab is a bispecific antibody. Aflibercept and conbercept (Lumitin; Cheogdu Kanghong Bio-Tech, Chengdu, China) are FcFPs. These anti-VEGF agents in nAMD treatment are typically administered through intravitreal injections, which results in vessel regression, reducing the size of the CNV area. There are anti-VEGF drugs approved for cancer treatment, including, for example, the VEGFR2 receptor inhibitor ramucirumab (Cyramza; Eli Lilly & Co, Indianapolis, IN, USA), tyrosine kinase inhibitors sorafenib (Nexavar; Bayer, Leverkusen, Germany and Onyx Pharmaceuticals, San Francisco, CA, USA), and sunitinib (Sutent, Pfizer, New York, NY, USA) [7]. It is also noteworthy that bevacizumab and aflibercept are approved drugs for cancer treatment. However, describing anti-VEGFs approved only for cancer treatment is beyond the scope of this review. We focus on anti-VEGF drugs currently used and approved for nAMD treatment here (Table 1).

### 3.1. Bevacizumab (Avastin, Genentech)

Bevacizumab was the first anti-VEGF drug approved for the treatment of metastatic colon carcinoma by the US Food and Drug Administration (FDA) in February 2004 [16] and by the European Medicines Agency (EMA) in January 2005 [17]. Bevacizumab binds with all isoforms of VEGFA and has been used off-label by medical clinicians for nAMD treatment due to its good efficiency and relatively cheap price. Specifically, in the United States, a single dose of bevacizumab costs approximately 40 times less than a single dose of ranibizumab [18]. However, for ocular use, bevacizumab should be repackaged into small doses under aseptic conditions because the original package is in a 4 mL (100 mg) or 16 mL (400 mg) vial at a concentration of 25 mg/mL. There are several biosimilars of Avastin that have been approved by the FDA and/or EMA [19,20]: Mvasi (Amgen, September 2017 FDA approved by FDA, January 2018 by EMA), Zirabev (Pfizer, February 2019 by EMA, June 2019 by FDA), Alymsys (Amneal pharmaceuticals, March 2021 by EMA, April 2022 by FDA), Vegzelma (Celltrion, August 2022 by EMA, Sep 2022 by FDA), Aybintio (Samsung bioepis, August 2020, approved by EMA), and Onbevzi (Samsung bioepis, November 2020 by EMA). Additionally, there are other biosimilars such as Avergra (Biocad, Russia), approved by the Russian Ministry of Health, and BAT1706 (Bio-Thera Solutions, China), approved by the China National Medical Products Administration.

### 3.2. Pegaptanib (Macugen, Bausch & Lomb)

Pegaptanib (Macugen; Bausch & Lomb, Laval, Canada) was originally developed by Gilead Science and licensed out to Eyetech/Pfizer and then to Bausch & Lomb. Pegaptanib is an oligonucleotide aptamer that specifically binds and inhibits VEGFA165. This now-discontinued drug, pegaptanib, was the first drug to be approved for nAMD treatment by the FDA in December 2004. However, pegaptanib, the specific VEGFA165 blocker, failed in the post-marketing stage due to the more efficient and effective bevacizumab and ranibizumab [21]. The insufficient effectiveness of the VEGFA165 blocker pegaptanib may be attributed to its exclusive inhibition of VEGFA165 and not any other isoforms of VEGFAs. Pegaptanib does not inhibit any other VEGFAs, even including the VEGF165b form, which is constitutionally very similar to the VEGFA165 [22] as mentioned above. In contrast, bevacizumab and ranibizumab are pan-blockers of VEGFAs. These indicate that more isoforms, as opposed to only VEGFA165, are involved in nAMD pathology and that the pathological mechanisms of VEGFs and VEGFRs in nAMD are more complicated than previously considered. Further, a considerable lesson is given to drug developers about drug specificity and the balance between safety and effectiveness/benefit. Of note, the current technology and knowledge might be able to update the aptamer approach even if pegaptanib fails.

### 3.3. Ranibizumab (Lucentis, Genentech)

Ranibizumab is an antigen-binding fragment (Fab) derived from bevacizumab (Figure 3), with a higher affinity for VEGFA compared to the parental bevacizumab Fab molecule [23]. It is further considered that the smaller size of the Fab, in contrast to a full antibody format, allows for better diffusion from the vitreous into the outer retina, and quicker clearance from systemic circulation, but with similar local ocular retention. A clinical comparison between ranibizumab (0.5 mg) and bevacizumab (1.25 mg) using the same monthly treatment protocol showed no difference in visual acuity (NCT00593450). It is also speculated that a ranibizumab treatment composed of smaller functional doses may be safer than bevacizumab, especially with the consideration of the data that there are higher rates of serious hospitalized systemic adverse events in bevacizumab-treated patients than those of ranibizumab patients [2]. Fab has no effector domain of antibody and no Fc receptor (FcRN) recycling, which indicates better clearance from the whole body system than full-size antibody drugs, and the functional concentration of Fab is almost half that of full-size antibody in the same molar concentration because Fab has one binding site within it, whereas antibody has two binding sites. Ranibizumab is currently used off-label for the treatment of retinopathy of prematurity (ROP). Byooviz (SB11, ranibizumab-nuna, Samsung Bioepis, South Korea) is the first biosimilar approved by the US FDA (September 2021) to treat nAMD [19,20]. Ranibizumab-eqrn Cimerli (Coherus Biosciences) is the second approval from the US FDA (August 2022). There are several biosimilars of rabibizumab, such as FYB201/CIMERLI^TM1^ (Formycon AG, German) [24], CKD-710 (Chong Kun Dang Pharmaceutical, South Korea) [25], R TPR 024 (Reliance Life Sciences, India), Xlucane (Xbrane, Sweden), and SJP-0133 (Kidswell Bio/Senju Pharmaceutical, Japan), on the track towards approval [19].

### 3.4. Aflibercept (Eylea, Regeneron Pharmaceuticals)

Aflibercept is a soluble chimeric recombinant FcFP consisting of VEGFR1 (domain 2), VEGFR2 (domain 3), and the Fc portion of IgG1. This VEGF trap, Eylea (2 mg), was approved in 2011 for the treatment of nAMD with an extended interval of 2 months [26]. A clinical trial comparing the monthly or every-2-month injection of aflibercept (0.5 mg and 2 mg) with the monthly injection of ranibizumab (0.5 mg) showed equivalent efficacy and safety among them [27]. Aflibercept exhibits a higher affinity with VEGFA than bevacizumab or ranibizumab and additionally targets VEGFB and PGF [8]. Aflibercept (0.4 mg) is currently the first anti-VEGF drug to be approved by the FDA (January 2023) for the ROP treatment [28], whereas ranibizumab is for off-label use for the ROP. A real-world meta-analysis between aflibercept and ranibizumab in the ROP treatment could/will provide valuable insights into aspects of efficacy and safety in the near future.

### 3.5. Brolucizumab (RTH258, Beovu, Novartis)

Brolucizumab was approved in 2019 for the treatment of nAMD with an extended drug treatment interval of up to 3 months [29]. Brolucizumab is a humanized rabbit anti-VEGF antibody fragment scFv in which variable fragment heavy chain (VH) and variable fragment light chain (VL) domains are linked together into one protein, forming an scFv. The small size (~26 kDa) and removed effector fragment crystallizable (Fc) region of brolucizumab are speculated to have greater potential for efficacy and safety due to its high rate of retinal penetration and high rate of clearance in serum (Figure 3). This speculation was proven in reality when the brolucizumab serum peak time showed 1–6 h, shorter than other anti-VEGF agents, whereas the ocular retention time was similar [30]. Clinical trials comparing the noninferiority of brolucizumab (6 mg) versus aflibercept (2 mg) showed comparable visual gains and superior anatomical outcomes with similar safety results [31,32,33]. However, in post-marketing surveillance, there were unexpected adverse events on the intraocular inflammatory spectrum, including retinal vasculitis and retinal vascular occlusion [34]. As a result, a warning and precaution regarding these potential events were added to the US FDA product label in 2020. A continued safety comparison study (NCT03710564) between brolucizumab (6 mg, every 4 weeks of administration) and aflibercept (2 mg, every 4 weeks of administration) revealed a higher risk of intraocular inflammation, including retinal vasculitis and retinal vascular occlusion, with rates of 9.3% vs. 4.5% for brolucizumab and aflibercept, respectively [35]. Patient blood samples experiencing ocular inflammatory adverse events disclosed high titers of anti-brolucizumab antibodies and exhibited robust T cell responses, indicating the immunogenicity of brolucizumab in a certain group of patients [36]. The currently licensed dose of brolucizumab is over 20 times greater than that of ranibizumab in molar concentration [30] (Figure 3). In consideration of doses and molar concentrations among other anti-VEGF drugs, using brolucizumab at the same or similar molar concentration may remove the adverse events or reduce the rate.

### 3.6. Conbercept (KH902, Lumitin, Chengdu Kanghong Biotechnology)

Conbercept is a soluble decoy/trap, similar to aflibercept, and a recombinant fusion protein of VEGFR1 (domain 2), VEGFR2 (domains 3, 4), and the Fc of IgG1. Compared to aflibercept, conbercept has an additional domain 4 of VEGFR2, which is alleged to enhance affinity and prolong the half-life of the molecule [37]. Conbercept binds to multiple VEGF members such as VEGFA, -B, and PGF as aflibercept does. Conbercept (Kd = 0.5 pM) [38] and aflibercept (Kd = 0.66 pM) [8] have similar binding affinities to VEGFA165 and have similar vitreous half-lives at 4.2 days [39] and 4.79 days [40], respectively, in rabbits. Conbercept was approved for nAMD treatment at extended 3-month interval injections by the China National Medical Products Administration in 2013. In clinical trials conducted in China, conbercept was injected at doses of 0.5 or 2.0 mg every month for 3 months, followed by injections every 3 months for 12 months, and then presented a visual acuity of 9- to 15-letter gains: 14.31 ± 17.07 letters (0.5 mg as necessary), 9.31 ± 10.98 letters (0.5 mg every month), 12.42 ± 16.39 letters (2.0 mg as necessary), and 15.43 ± 14.70 letters (2.0 mg every month) compared to the baseline [37]. So far, more than 0.36 million patients in China have received conbercept treatment, although the published pharmacodynamic data of conbercept in the eyeballs and serums of nAMD patients after treatment are still insufficient. A global phase 3 clinical trial for conbercept (NCT03577899) was projected for its comparison with aflibercept in more than 300 sites across 30 countries in September 2018. However, the global clinical trial was dropped in 2020 due to the COVID-19 pandemic, which resulted in a large number of patients discontinuing their treatment and missing follow-up. There are several reports on meta-analyses of clinical observations of conbercept treatment with different results [41,42], so the properly designed phase 3 clinical data publication about the safety and efficacy of conbercept treatment will be critical and impactful for all those in this sector of the scientific and medical communities.

### 3.7. Faricimab (Vabysmo, Genentech)

Faricimab is the most recent drug approved for nAMD treatment by the US FDA in January 2022 (Table 1). Faricimab is a bispecific monoclonal antibody that blocks VEGFA and angiopoietin-2 (ANG2) (Figure 3). Faricimab has an inhibitory effect on both VEGFA and ANG2, with extended treatment intervals of up to 3 months [43]. Faricimab has been shown to be non-inferior to ranibizumab (phase 2) and aflibercept with up to 4-month interval treatment protocols (phase 3) [44,45,46,47]. Faricimab showed a similar rate of serious ocular adverse events to aflibercept and no evidence of occlusive retinal vasculitis [47,48]. ANG2 is one of the pro-angiogenic molecules elevated in the vitreous of patients with retinal vascular diseases [49,50,51]. ANG2 binds to the TIE2 transmembrane receptor tyrosine kinase and competes with ANG1. ANG2 and ANG1 have different functions with the same receptor, TIE2 binding. The binding of ANG2/TIE2 decreases vascular stabilization, whereas ANG1/TIE2 increases vascular stability and inhibits permeability. The coexpression of ANG2 and VEGFA in ischemic retina accelerated neovascularization [52], and the combinatory inhibition of VEGFA and ANG2 more aggressively reduced vessel lesions compared to the single treatment in CNV mice and laser-induced non-human primate models [51]. In addition, the FcRN and FcγRs binding of faricimab is disabled to inhibit Fc-medicated effector function and FcRN recycling, endowing better safety and systemic clearance [51]. Among the anti-VEGF therapeutics so far, faricimab is considered the most effective and safest for nAMD treatment. However, it will be meaningful to mention that vanucizumab (ANG2-VEGFA Cross Mab, Roche) failed to show improved clinical outcomes in metastatic colorectal cancer in comparison with bevacizumab [53], although ANG2-VEGFA Cross Mab showed a strong inhibition of angiogenesis and tumor regression in animal tumor model mice [54].

### 3.8. Ranibizumab Ocular Implant (Susvimo, Genentech)

Susvimo, a port delivery implant with ranibizumab, was approved by the US FDA in October 2021 for the nAMD treatment. Susvimo is a ranibizumab delivery system that consists of a silicon shell containing a drug reservoir that is implanted into the eye through a sclera and pars plana incision and continuously releases ranibizumab over a period of 6 months in the implanted eye [55,56]. The susvimo implant system is non-degradable and is refilled every 24 weeks. At the phase 3 clinical trial, susvimo (every 24 weeks) was non-inferior to ranibizumab injection (0.5 mg every month) in gaining visual acuity within 9 months [57]. However, susvimo prescribing information warns of a threefold higher rate of endophthalmitis than the monthly IVT of ranibizumab, as well as conjunctival erosion and retraction.

## 4. Future Anti-VEGF Drugs

Overall, anti-VEGF drugs have demonstrated their successes, and we have had a variety of anti-VEGF antibody-derived drugs. Additionally, patients have had the benefits of anti-VEGF treatment for the last 20 years in the treatment of nAMD. Ocular pathological neovascularization is no longer a non-manageable disease. However, there is still room for improvement in anti-VEGF drugs, and several candidates for anti-VEGF agents are under clinical trials.

For one, the recent bispecific antibody drug, faricimab, is quite successful in inhibiting both molecules of VEGFA and ANG2 in one antibody. This approach can control and inhibit molecules in different pathways associated with pathological conditions. Efdamrofusp alfa (IBI302; Innovent Biologics, Suzhou, China) is a bispecific fusion protein that targets both VEGF and C3b/C4b, and IBI302 is under phase 2 clinical trials (NCT04820452, NCT05403749). The VEGF inhibitor domain of IBI302 is similar to that of aflibercept, and the C3b/C4b inhibitor domain is derived from complement receptor 1.

Another approach is to improve the delivery system by extending its duration. The recently approved susvimo is a great innovation, with a non-degradable port and drug refill system. Gene delivery via viruses is another approach. Gene delivery via viruses is a partly long-lasting drug approach to reduce patients’ burden of repeated injections, although there are still concerns about the safety of immunogenicity and virus genomic integration. RGX314 (AAV8-anti-VEGF Fab; Regenxbio, Gaithersburg, MD) is under phase 2, 2/3, or 3 of clinical trials by subretinal administration (NCT03999801, NCT04832724, NCT04704921, NCT05407636) or suprachoroidal space administration (NCT05210803, NCT04514653), in comparison with the effects of no-intervention, ranibizumab, or aflibercept. AAV2-IBI302 (bispecific anti-VEGF and anti-C3b/C4b FcFP, Innovent Biologics) has also shown good efficacy in reducing retinal inflammation and neovascularization in uveitis and CNV animal models [58].

Furthermore, targeted protein degradation technologies hold promises for future applications. The first translational study for extracellular VEGF degradation is bi-AbCap (Eli Lilly & Co) [59]. The bi-AbCap is a fusion antibody system consisting of an anti-insulin growth factor type I receptor (IGF-IR) antibody and VEGFR1 (domain 2). The anti-IGF-IR and anti-VEGF bi-AbCap were designed to capture extracellular VEGF, internalize VEGF through IGF-IR, and finally degrade captured VEGF using intracellular lysosome. This targeted degradation technique of bi-AbCap was demonstrated in a cancer cell model, but it is applicable to other diseases, including nAMD. Targeted protein degradation strategies have been mostly limited to cytoplasmic proteins. However, there are a few technologies for targeted extracellular protein degradation through receptor-mediated endo-lysosomal clearance, including bi-AbCap, as mentioned above: lysosome-targeting chimeras (LYTAC), cytokine receptor-targeted chimeras (KlineTacs), and asialoglucoprotein receptor (ASGPR) chimeras [59,60,61,62]. LYTAC consists of a glycoprotein ligand and an extracellular protein-binding part. The glycoprotein ligand of LYTAC activates the cation-independent mannose-6-phosphate receptor, and the ligand and receptor complex are transported into lysosomes [60]. KlineTac is a bispecific antibody consisting of a cytokine arm and a target-binding arm for extracellularly interesting proteins. KlineTac adopts a natural cytokine degradation mechanism [61]. ASGPR chimeras consist of small-molecule ASGPR ligands and target protein binders [62,63,64]. In all these receptor-mediated targeted protein degradation techniques, extracellular target proteins are arrested by membrane receptor-targeted chimera-binding capture systems, internalized via receptor recycling mediation, transported to lysosomes, and finally degraded.

## 5. Concluding Remarks

The historical review of anti-VEGF drug development for the treatment of nAMD is inspiring and educational for translational researchers working in any antibody drug development area. Drug development evolution, such as the multiple target approach, just opened a new era of antibody drugs, including anti-VEGF antibodies. Furthermore, nucleotide synthetic technology easily improves the binding affinity, stability, and specificity of antibody drugs. When we understand more about the molecular and pathophysiological aspects of nAMD, we could approach various combinatory drug development strategies for bispecific or even triple-specific agents in one antibody drug. Finally, recent progress in gene delivery and targeted extracellular protein degradation techniques may lead to greater evolution in efficacy, efficiency, and safety in the near future and, finally, improvement in the quality of life and economic burden of patients.

## Figures and Tables

**Figure 1 ijms-24-13004-f001:**
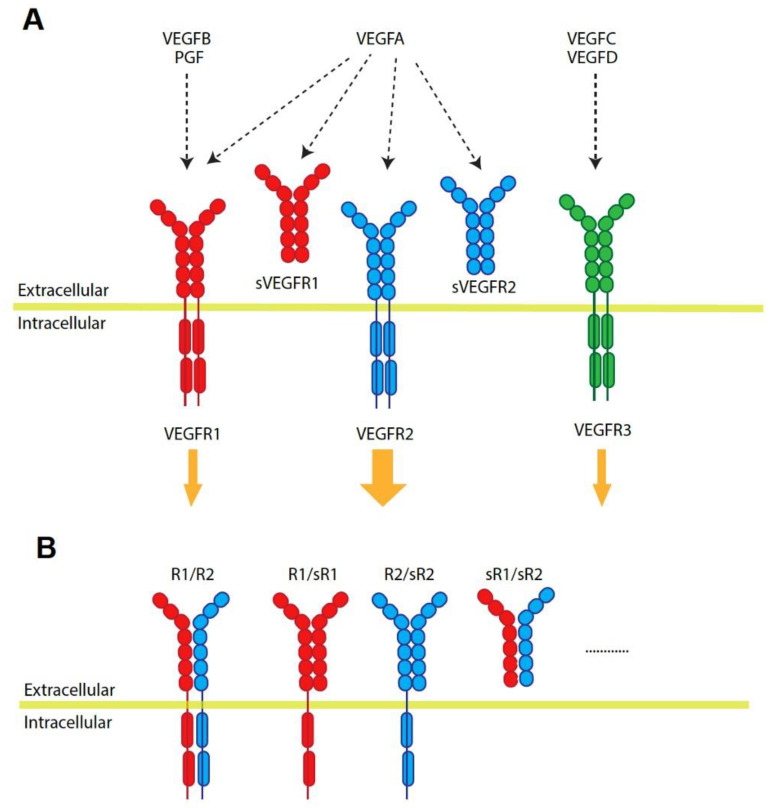
Vascular endothelial growth factor (VEGF) and VEGF receptor (VEGFR) interaction. (**A**) VEGFB and placental growth factor (PGF) bind to VEGFR1. VEGFA binds both VEGFR1 and VEGFR2. VEGFC and VEGFD bind to VEGFR3. VEGFR2 activation shows the strongest tyrosine phosphorylation and intracellular signaling. The binding of VEGFs and soluble VEGFR1 and R2 exerts an inhibitory effect. (**B**) The homodimerization of VEGFRs is a major format, but VEGFRs create heterodimers and even dimerize with soluble ones.

**Figure 2 ijms-24-13004-f002:**
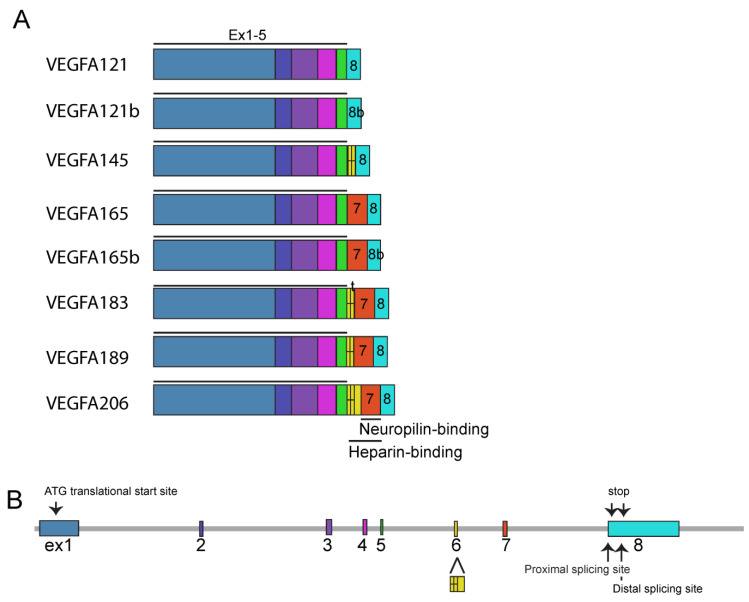
Isoforms of human vascular endothelial growth factor A (VEGFA). (**A**) VEGFAs are composed of eight exons and subtypes with different lengths such as VEGFA121, VEGFA165, and VEGFA206. All subtypes contain exons 1–5. Alternate splice site selection in exons 6–8 generates multiple isoforms. Exons 6 and 7 encode heparin- and neuropilin-binding domains. Exon 8 can be spliced at a proximal or distal splicing site. Distal splicing generates the VEGFAxxxb form with a different C-terminus (indicated 8b). ^t^, truncated form. (**B**) Exon structure of the VEGFA gene. Size is adopted from UCSC Genome Browser.

**Figure 3 ijms-24-13004-f003:**
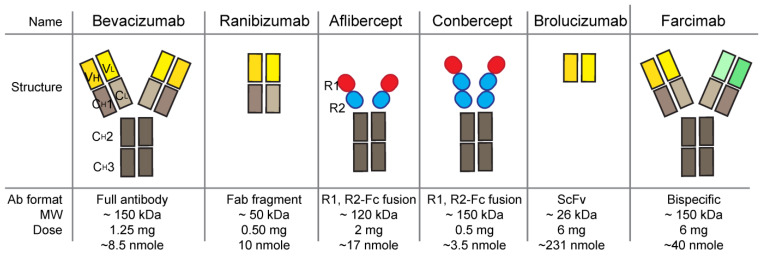
Illustrative structure of anti-VEGF antibody-derived drugs. Molecular weights, clinical doses, and approximate molar amounts of clinical doses are summarized. Region of anti VEGF is indicated by yellow domains and region of anti-ANG2 is indicated by green. V_H_, variable region of heavy chain; V_L_, variable region of light chain; C_H_1, constant region 1 of heavy chain; C_H_2, constant region 2 of heavy chain; C_H_3, constant region 3 of heavy chain; C_L_, constant region of light chain.

**Table 1 ijms-24-13004-t001:** Summary of anti-VEGF drugs.

Name	Commercial Name	Sponsor	Approved Year	Type	MOA	Doses	Interval
Bevacizumab	Avastin	Genentech	Off label	mAb	VEGFA inhibition	1.25 mg/0.05 mL	Q1M
Pegaptanib	Macugen	Bausch/Lomb	2004	aptamer	VEGFA165 inhibition	0.3 mg/0.09 mL	Q6W
Ranibizumab	Lucentis	Genentech	2006	Fab	VEGFA inhibition	0.5 mg/0.05 mL	Q1M
Aflibercept	Eylea	Regeneron	2011	FcFP	VEGFA, VEGFB, and PGF inhibition	2 mg/0.05 mL	Q1M for 3M and then Q2M
Conbercept	Lumitin	Cheongdu Kanghong Biotech	2013 (China)	FcFP	VEGFA, VEGFB, and PGF inhibition	0.5 mg/0.05 mL	Q1M for 3M and then Q3M
Brolucizumab	Beovu	Norvatis	2019	scFv	VEGFA inhibition	6 mg/0.05 mL	Q1M for 3M and then Q2-3M
Ranibizumab ocular implant	Susvimo	Genentech	2021	mAb implnat	VEGFA inhibition	2 mg/0.02 mL, continuously released for 6 months	Refill Q24W
Faricimab	Vabysmo	Genentech	2022	mAb	VEGFA and Ang2 inhibition	6 mg/0.05 mL	Q1M for 4M and then Q2-3M

Approval year is based on US FDA approval, if not indicated. Q, Latin quaque, means each; Q1M means once monthly.

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
