# Peer review of "Twenty Years of Anti-Vascular Endothelial Growth Factor Therapeutics in Neovascular Age-Related Macular Degeneration Treatment"

_ijms, 2023, doi:10.3390/ijms241613004_

Round 1

Reviewer 1 Report

The manuscript is well-written and informative, and gives a comprehensive review on structures, functions, and clinical applications of various anti-VEGF drugs in nAMD treatment for the past 20 years. Among the eight anti-VEGF drugs being discussed in the paper, Pegaptanib is less common and has been discontinued in the US. I wonder could this part be move to the end of the discussion in “3. Molecular and clinical aspects of anti-VEGF drugs”.

Minor points:

1.     Please label each item in figure 1B.

2.     Please spell out the full name of ATG, PSS and DSS in figure 2B

3.     Table1 bebacizumab should be bevacizumab

4.     The title of Table 1 is not concise, please consider revise it

Author Response

We reply to the reviewer 1:

The manuscript is well-written and informative, and gives a comprehensive review on structures, functions, and clinical applications of various anti-VEGF drugs in nAMD treatment for the past 20 years. Among the eight anti-VEGF drugs being discussed in the paper, Pegaptanib is less common and has been discontinued in the US. I wonder could this part be move to the end of the discussion in “3. Molecular and clinical aspects of anti-VEGF drugs”.

We have considered removing “Pegaptanib” part from the section of “molecular and clinical aspects of anti-VEGF drugs”, but we decided that we would keep it because we hoped to give whole historical view of all approved drugs including post-market failed one. Basically we think that more specific drugs are safer, but we should think the balance between safety and effectiveness. At current, antibody drugs have been quite successful in this area, but still there is room for nucleotide drug development with more updated techniques because we failed. We additionally mention our view in the part of Pegaptanib:

lines 157-160, Further, additional considerable lesson is given to drug developers about the drug specificity and balance between safety and effectiveness/benefit. Of note, current technology and knowledge might be able to update aptamer approach even if pegaptanib failed.

Minor points:

  1. Please label each item in figure 1B.

-> Each item in figure 1B was labelled.

  1. Please spell out the full name of ATG, PSS and DSS in figure 2B

      ->  We spelled out the exact or full names of all of them in figure 2B

  1. Table1 bebacizumab should be bevacizumab

->  Word was corrected. Thanks

  1. The title of Table 1 is not concise, please consider revise

-> Title was concisely revised: Summary of Anti-VEGF drugs approved except biosimilars à Summary of approved anti-VEGF drugs.

Reviewer 2 Report

Well-prepared and well-organized paper. This is a brief review about a delicate topic involving vision, aging, clinics, studies over last twenty years. Research has always been a lot, therapeutic options for wet-AMD are getting more specific but unfortunately not flawless. This manuscript takes into account all current available drugs against exudative AMD, but it can sum up the topic in an effective way, without being long and boring. Figures are intelligible and very clear. The reference list is updated as much as possible, the wide area of the subject notwithstanding.

Author Response

Thank you for the comments.

Minorly, we have updated:

  • In figure 1B, we labeled each item in figure 1B
  • In figure 2, abbreviated words were given as full names
  • Line 9, remove space before starting sentence
  • Line 29, change fonts
  • Line 99, human vascular endothelial growth factor A à human vascular endothelial growth factor A (VEGFA)
  • Line 124, Summary of Anti-VEGF drugs approved except biosimilars à summary of approved anti-VEGF drugs
  • Table1) Bebavizumab -> Bevacizumab
  • Table 1) added: Q, Latin quaque means each: Q1M means once monthly
  • Line 266, keep font symbol - FcgRs

Reviewer 3 Report

This paper summarizes the available monoclonal antibodies, which are now considered for treatment of AMD. It is an important review on an ophthalmological disorder that frequency is high and has only limited possibilities for treatment at present.

Due to of these, this referee suggests the authors to add an additional section describing the pathology of AMD.

Some minor typos have been detected, please correct:

Introduction, line 5: add vascular endothelial growth factor before abbreviation VEGF

Fig. 2. legend line 1: ...A (VEGFA).

Table 1. Bevacizumab

I consider the manuscript suitable for publication in IJMS.

Minor editing of English language required

Author Response

We carefully considered adding another section, but we decided to provide more background in introduction part instead of adding another section. Please check lines 33-41.

Lines 33-41: there are Age Related Eye Disease Study (AREDS), dietary supplements for the early and intermediate AMD, anti-complement pathway drugs of pegcetacoplan (Syfovre, c3 inhibitor, Appellis) and avacincaptad (Izervay, IVERIC) for GA, and several anti-vascular endothelial growth factor (VEGF) drugs such as bevacizumab (Avastin, Genentech), ranibizumab (Lucentis, Genentech), and aflibercept (Eylea, Regeneron), in the treatment of neovascular AMD (nAMD), CNV. Whereas anti-complement pathway drugs were newly approved in 2022 and 2023 to treat GA, and still require real-world post-market proof of the effectiveness, anti-VEGF drugs have shown already quite successful efficacy for the last 20 years

Some minor typos and errors were corrected:

  • Line9, remove space before starting sentence
  • Line29, change fonts
  • Line99, in fig2 legend, human vascular endothelial growth factor A -> human vascular endothelial growth factor A (VEGFA)
  • Line124, Summary of Anti-VEGF drugs approved except biosimilars -> summary of approved anti-VEGF drugs
  • Table1) Bebavizumab -> Bevacizumab
  • Table 1) added: Q, latin quaque means each: Q1M means once monthly
  • Line 266, keep font symbol